# Health Community 4.0: An Innovative Multidisciplinary Solution for Tailored Healthcare Assistance Management

**DOI:** 10.3390/s24186059

**Published:** 2024-09-19

**Authors:** Sara Jayousi, Chiara Barchielli, Marco Alaimo, Sara Guarducci, Stefano Caputo, Marzia Paffetti, Paolo Zoppi, Lorenzo Mucchi

**Affiliations:** 1PIN—Polo Universitario “Città di Prato”, 59100 Prato, Italy; sara.jayousi@pin.unifi.it; 2Management and Health Laboratory, Institute of Management, Sant’Anna School of Advanced Studies of Pisa, 56127 Pisa, Italy; chiara.barchielli@santannapisa.it; 3Department of Nursing and Midwifery, Local Health Unit Toscana Centro, 50139 Florence, Italy; marco.alaimo@uslcentro.toscana.it (M.A.); marzia.paffetti@uslcentro.toscana.it (M.P.); paolo.zoppi@uslcentro.toscana.it (P.Z.); 4Department of Information Engineering, University of Florence, 50121 Florence, Italy; sara.guarducci@unifi.it (S.G.); stefano.caputo@unifi.it (S.C.)

**Keywords:** information and communication technologies, healthcare, nursing, patients’ management, Internet of Things, health monitoring

## Abstract

This paper presents a co-designed solution aimed at improving community healthcare assistance management. To enhance patients’ self-awareness of their health status, encouraging active participation in care process, and to support nurses in delivering patient-centered care, we have developed an innovative platform with a highly customized app. This platform was designed using a multi-disciplinary, bottom-up approach. Patient data collection and processing facilitate the automation of care timeline planning, based on real-time patients’ needs and the available resources. To achieve this goal, different components have been considered: real-time health data collection and processing, patient care planning, decision support for nurses, secure communication for data transmission, and user-friendly interfaces to ensure easy access to platform functionalities.

## 1. Introduction

### 1.1. Background and Motivation

A chronic health condition poses a significant challenge in a patient’s life. Managing it is often difficult, time-consuming, and not always effective, especially since many patients suffer from more than one chronic health issue. Addressing these challenges in community healthcare management requires strategic methodologies that facilitate a personalized care approach for several key reasons. Individuals have diverse healthcare needs, cultural backgrounds, and preferences, necessitating individualized solutions. Furthermore, community healthcare programs frequently encounter limited resources, making efficiency imperative. Additionally, tailored and efficient interventions are more likely to achieve the desired results. Patients who receive personalized care tend to be more involved in their treatment plans, experience better health outcomes, and generally require less intensive interventions [1]. Furthermore, those who perceive their needs as fully met are more likely to develop greater trust and engagement with the healthcare system [2,3]. Consequently, it is crucial to explore how healthcare systems can maintain sustainable practices while ensuring comprehensive coverage and high-quality care services that reflect a patient-directed approach. Indeed, the efficient management of patients, especially those with chronic health conditions, through a well-planned community-based assistance, is essential for both current and future healthcare scenarios.

The integration of Information and Communication Technology (ICT) capabilities with implemented care models allows for the definition of dynamic and adaptive services that meet the needs of citizens and patients while supporting healthcare personnel in managing an increasing number of patients through a 4.0 territorial assistance model. Traditional healthcare models have predominantly been reactive, focusing on treatment after the onset of disease [4]. However, with the rise of digital health technologies, there is a significant shift toward proactive and preventive approaches that can lead to early detection of potential health issues, allowing for prompt and appropriate actions [5,6]. Health community 4.0 embodies this shift, leveraging real-time data and advanced analytics to predict and prevent health issues before they become critical. Furthermore, this type of management, in addition to requiring advanced technologies such as telehealth platforms, remote monitoring devices, and wearable health sensors, also necessitates a well-defined strategy, which we believe should be a bottom-up process.

Healthcare systems worldwide have traditionally employed a top-down approach, in which large institutions take the lead by allocating resources and centralizing efforts for extensive projects. While this method can foster substantial change, its impact is often short-lived. A significant limitation of this strategy is its tendency to generate minimal engagement and a lack of ownership among front-line staff, who are critical to the success of any healthcare initiative. Furthermore, by failing to address the underlying behaviors and unique needs of the community, this approach often results in the long-term unsustainability of projects. Thus, a “one-size-fits-all” approach is widely recognized as inadequate in addressing the complexity and diversity of individuals’ needs, resulting in suboptimal care and, in many cases, excluding people from accessing essential services [7,8,9]. Recently, there has been a notable shift toward adopting bottom-up strategies in public healthcare management, although some models still integrate elements of both top-down and bottom-up approaches [10,11,12]. This trend is driven by a growing recognition of the need for more inclusive and participatory systems [13,14,15,16,17,18,19,20]. Bottom-up approaches, which are initiated at the grassroots level, place a strong emphasis on involving and leveraging individual contributions to construct a more comprehensive and complex system [21]. Unlike the top-down approach, which relies on centralized authority, the bottom-up model promotes a co-managed care framework. This method empowers healthcare providers, patients, and families by fostering inclusivity and allowing diverse perspectives and ideas to emerge organically. By encouraging collaboration and innovation, this approach ensures that users actively participate in the design and development of healthcare systems and services [22]. Evangelista and colleagues, for instance, explored the implementation of a remote monitoring system based on the principles of community-based participatory research to manage chronic heart failure in older adults. They engaged patients representative of the target population to collaborate with the research team in designing the proposed health platform [13]. A similar methodology was later employed by Hochstenbach’s group to develop a co-creative e-health intervention aimed at supporting self-management in cancer outpatients experiencing pain [14]. A co-design approach, which actively engages end-users throughout the product development process, was also applied to create new digital health interventions for renal patients [16] and esophageal cancer patients [20], in some cases integrating advanced wearable sensors [18]. Some studies have also adopted a bottom-up approach to developing e-health systems supporting healthcare professionals in nursing homes [15,17].

Despite the recent increasing use of user-centered design within healthcare, the application of this approach for managing community healthcare assistance, particularly with home-visit nurses as target end-users, remains unexplored. Therefore, the approach we propose aligns with the emerging trend of bottom-up strategies, which we believe holds significant potential to transform the organization and management of healthcare systems at the community level.

When implementing a participative approach, it is essential to consider its adoption among end-users. As stated by the Unified Theory of Acceptance and Use of Technology (UTAUT) [23], the success of the developed solution relies on active participation and sustained use by the communities involved in its creation [24]. UTAUT particularly identifies social influence and facilitating conditions as significant factors influencing technology adoption. Social influence represents perceived pressure from others to use technology, which, in the context of community participation in solution creation, can foster ownership and peer support. Facilitating conditions, including resource availability, technical support such as tailored training and ongoing assistance, and clear communication channels, are essential for ensuring that users feel competent and comfortable using the solution effectively.

This methodology represents a transition from traditional to modern medicine through the concept of “super convergence”. Modern medicine integrates elements such as social networking, wearable and environmental sensors, and computational power, forming the basis of a new healthcare paradigm centered on advanced technologies and aimed at enhancing patient care. The evolution toward modern medicine recognizes patients as health consumers, emphasizes the importance of home as the primary place for care, and ensures no one is left behind. Patients are more likely to feel supported and understood, knowing that their care is being systematically managed and that they are active participants in their healthcare journey [25]. Consequently, this streamlining of processes allows for better coordination of care, timely interventions, and personalized treatment plans, leading to increased patient satisfaction [26,27]. Education and regulatory frameworks are also essential for ensuring that both patients and healthcare providers are well-informed and compliant with best practices. Guiding principles include homogeneity, balance, and equity alongside an integrated approach to health pathways and adaptable resource management. Also, fostering a culture of innovation is pivotal for ongoing quality improvement in healthcare. This approach, centered on occurrences and specific needs of patients and caregivers, not only enhances preventive healthcare measures but also significantly contributes to optimizing community healthcare assistance overall [27,28,29].

### 1.2. Our Contribution

Recently, healthcare management has been undergoing a transformative evolution with the integration of self-monitoring, monitoring, and patient journey tracking into a unified system [30]. By utilizing real-time data, personalized care plans, and continuous feedback, healthcare providers can significantly enhance patient outcomes, improve operational efficiency, and deliver high-quality, cost-effective care. As healthcare continues to evolve, the adoption of such innovative solutions is essential in meeting the increasing demands and expectations of both patients and providers.

Our work has focused on emphasizing the need for a tool that integrates patient self-monitoring, professional monitoring, and patient care planning. This tool is designed not only to enhance patient management but also to improve actual organizational models based on collected feedback, such as deviations from the planned outcomes. Specifically, we proposed a multidisciplinary bottom-up approach named smartHUB, which emphasizes end-user engagement from the initial stages and throughout the design process. The aim is to implement a 4.0 assistance system capable of effectively addressing future challenges related to citizen health and well-being. This innovative approach has been developed within the framework of the smartHUB project and has been applied to the design and development of new healthcare assistance tools for the Local Health Unit (LHU) Toscana Centro, a component of the Tuscany Health Service known for its highly advanced healthcare system [31]. In 2018, Tuscany pioneered the implementation of the Family and Community Nurse (FCN) model with the approval of DGR 597/18. Under this model, FCNs are designated as the primary professionals responsible for delivering nursing care across various levels of complexity. They collaborate with all community professionals, not only providing direct patient care but also engaging with all relevant stakeholders and resources to address both current and anticipated needs within the community. To address the variety of territorial needs, we propose a scalable pathway that can be carefully customized as needed, having the potential for expansion with new functionalities and ensuring interoperability with existing systems. Furthermore, this system is designed to adapt to different healthcare settings beyond LHU Toscana Centro, including the broader European context.

The identified ICT solution, designed to effectively enhance territorial care, consists of an innovative platform that includes a highly tailored mobile app. Such a system is specifically planned for health data collection through patient self-monitoring or during home visits by nurses, with the aim of increasing patients’ self-awareness regarding their health status and supporting nurses in the provision of patient-centered care. One of the primary goals of this initiative is to streamline healthcare processes and provide comprehensive support to patients, thereby fostering a more connected and efficient healthcare environment.

The paper is organized as follows: After providing the background context and the motivation behind the need for innovative ICT tools for effective healthcare management (Section 1), in Section 2 we describe our multidisciplinary approach called smartHUB, focusing on how it has been conceived to stimulate and support local administrations, social and health institutions, and third sector organizations. Section 3 outlines the co-designed creation of the system, from users’ needs to system implementation, while Section 4 offers an overview of the architecture of the proposed solution, the PROASSIST 4.0 system. This is followed by a detailed description of the developed mobile application, the PROASSIST 4.0 app, including its profiles and functionalities (Section 5). Furthermore, Section 6 highlights the refinement of our app based on initial findings from the system evaluation phase, in addition to detailing upcoming experimental activities intended to validate these app adjustments. Finally, Section 7 discusses the presented solution and associated challenges, concluding with remarks in Section 8.

## 2. The smartHUB Approach: An Innovative Multidisciplinary Systemic Methodology

The smartHUB methodology employs a systemic management approach to innovation processes, engaging all stakeholders (users, service providers, companies, researchers, and policymakers) from the initial needs analysis through the entire design process.

It consists of four macro-phases (Figure 1), which are briefly described in the following subsections:-Macro-phase 1: Data Collection and Input;-Macro-phase 2: Analysis, Co-Design, and Development;-Macro-phase 3: Validation Experiments;-Macro-phase 4: Implementation.

### 2.1. Macro-Phase 1: Data Collection and Input

The initial macro-phase focuses on data collection, beginning with active community involvement. After determining the application context, the critical step of “listening to the territory’s needs” helps define the final technological solution. Essential participants in social and healthcare service delivery, along with their beneficiaries, play an active role in identifying both current and emerging needs. Moreover, community engagement offers a detailed overview of the implemented solutions, highlighting their advantages and disadvantages concerning operational activities.

Besides performing a preliminary analysis of user needs, existing solutions, and stakeholder feedback, this macro-phase also considers the state-of-the-art in technological research. This is to explore both current and emerging technologies that could be adopted to create innovative solutions addressing the identified needs. Data collection methods employed during this macro-phase include workshops, focus groups, and literature reviews.

### 2.2. Macro-Phase 2: Analysis, Co-Design, and Development

The second macro-phase is composed of three steps: (i) co-design and co-creation; (ii) identification of innovative solutions for coordination, integration, and training actions; (iii) detailed design and development.

The co-design and co-creation phase involves a thorough analysis of the gathered data, with a particular focus on evaluating existing solutions and their potential and limitations. This step aims to identify gaps in relation to current and anticipated technological advancements.

By employing a multidisciplinary co-design and co-creation approach that involves frequent interactions with end-users, innovative solutions are identified. Additionally, this approach facilitates the development of coordination, integration, and training actions to enable active experimentation within the community.

The final step of this macro-phase involves the realization of the identified solution. This includes translating the high-level project into a detailed design by defining system requirements and functional components necessary for development and integration. Furthermore, this step encompasses the execution of functionality, integration, and internal validation tests, preparing the solution for real-world testing with end-users.

### 2.3. Macro-Phase 3: Validation Experiments

The third macro-phase serves as a small-scale experimental validation campaign conducted within the community. Its aim is to achieve preliminary validation and evaluation of the proposed solution in response to emerging needs, through the incremental refinement of the system. This phase is crucial for implementing and standardizing innovative solutions and actions that provide significant value to individual well-being and health, which everyone should be aware of.

This macro-phase comprises two main steps: (i) experimental and evaluation activities and (ii) tuning. Experimental activities involve testing the proposed solution in a controlled environment with active participation from end-users, gathering their feedback through interviews and specially designed questionnaires. The tuning step involves adjusting the solution based on the feedback analysis. As illustrated in Figure 1, this step may require detailed design revisions and/or the addition of new features to enhance the solution. The feedback–tuning cycle may need multiple iterations, depending on the extent of revisions necessary.

### 2.4. Macro-Phase 4: Implementations

The fourth macro-phase involves implementing the solution within the community, thus concluding one operational cycle and initiating the next. Evolving societal needs and the introduction of innovative solutions can lead to new requirements and changes. Therefore, continuously “listening to the community” is crucial for defining technological systems that effectively support social and healthcare services.

### 2.5. The smartHUB Approach in the Healthcare Assistance Management Framework

In the healthcare assistance management context, the smartHUB approach has the ultimate goal of improving patient care and quality of life [28], linking the digital world with the real one to benefit the patient [29]. One of the primary objectives of this initiative is to streamline healthcare processes and offer comprehensive support to patients, fostering a more connected and efficient healthcare environment. Co-design and development phases focus on user-centered principles, creating an intuitive and accessible interface suitable for patients of all ages and technical abilities.

## 3. The Co-Designed PROASSIST System: From Users’ Needs to System Implementation

### 3.1. Application Context: Nursing Home Assistance

The smartHUB approach has been adopted to design and develop effective ICT tools for improving community healthcare nursing assistance in LHU Toscana Centro. In particular, the process starting with gathering users’ needs has led to the implementation of the PROASSIST 4.0 system, that will be tested, validated, and evaluated in a real context. In Figure 2, an overview of the main steps needed for the system co-design and implementation is provided. These steps are mapped into the smartHUB approach macro-phases (figure left side) and into the co-design workshops held with users (figure right side). We organized four workshops with all end-users (FCNs, nurse coordinators, and patients associations) in an effort to analyze the specific needs of territorial nursing care and provide the initial definition of possible improvement strategies for community healthcare management. During each workshop, particular attention was given to the areas of patient care, health monitoring, care pathways and emerging needs. Specifically, the first three sessions (co-design and co-creation workshops) were intended to identify the main actors of territorial nursing care and collect information about the current state of the art of community healthcare assistance, users’ emerging needs, and nurses’ desires related to their work management. Data gathered during the co-design and co-creation activities were then analyzed to define the main user requirements and the most relevant use cases for the solution design. Finally, the last workshop was oriented toward exposing and evaluating the implemented ICT solution resulting from the previous sessions in order to collect users’ feedback essential for system tuning. Several processes for system tuning have been conceived and implemented to develop a ICT tool that is functional and effective for the community healthcare nursing services at LHU Toscana Centro. Nevertheless, the proposed solution remains under continuous development, and ever more specific improvements will be required to achieve the final implementation.

### 3.2. Users’ Needs: Identification and Classification

Users’ emerging needs we collected during the co-design and co-creation workshops can be grouped in four main categories:Territorial needs;Resource needs;Communication needs;Barrier needs.

Within each category, for each specific need, the desired outcomes (how one would wish it to be) were solicited. Based on these inputs, a proposal of functionalities was formulated to satisfy the users’ requirements, always within the specific operational context of reference.

#### 3.2.1. Territorial Needs

Territorial needs arise from the observation that each territory has its own specificities, with central and peripheral zones differing in terms of ease of reaching the patient. Accessing peripheral areas entails a longer time, while in the urban areas, despite distances being limited, traffic represents a significant concern. Territorial needs also deal with the absence of cellular network coverage in some peripheral areas and with the lack of knowledge regarding the services available within the territory. To address these emerging needs, the following functionalities have been proposed: (i) location-independent connectivity (e.g., alternative networks); (ii) traffic-based route optimization; and (iii) territorial maps with indications of gathering points and local services.

#### 3.2.2. Resource Needs

Resource needs encompass shortages in personnel, such as a lack of specialized medical staff, as well as deficiencies in technological tools used for management and organization of care programs. Currently, health parameter measurement and data collection during home visits are manually conducted on paper, with subsequent transcription into the corporate program. In addition, there is a lack of accessible historical patient data and technological tools enabling personalized weekly planning and management of criticalities in patient care. In order to comprehensively respond to these needs, we proposed the development and implementation of the following functionalities: (i) direct input of health data into the automated system (via connected devices) or through manual entry or voice assistant; (ii) patient data history with graphical reports, shared among all healthcare professionals involved in patient care; (iii) a patient-specific approach for accurate data interpretation and goal assessment; (iv) automated planning as an initial recommendation for management of patient care; (v) proximity-based nurse localization for patients with critical conditions; (vi) personalized data correlation algorithms with priority indication; (vii) tracking and reminders to ensure therapy adherence; and (viii) color notifications to patients for anxiety mitigation and reduced unnecessary communication.

#### 3.2.3. Communication Needs

Communication needs arise from the necessity of a more efficient interaction among all healthcare professionals involved in patient care, with the aim of facilitating the management of patient’s health status, particularly in critical conditions. Among communicative needs, there is also the necessity to establish a visual interaction between healthcare workers and the patient, essential for better understanding patient’s non-verbal behaviors and building a relationship based on empathy. Potential functionalities that could support healthcare professionals in their work include: (i) real-time communication for effective collaboration among all actors involved and (ii) video calls for direct interactions with the patient and visual management of medical conditions. Proposed functionalities that could help patients consist of: (i) an opportunity for patients to access to their health data in a modality appropriate for them, fostering their involvement and awareness of their treatment journey while mitigating potential anxiety; (ii) the possibility for patients to know that nurses are effectively managing their health situation in order to perceive to be actively monitored; and (iii) reminders for therapy, healthy lifestyle practices, and appointments.

#### 3.2.4. Barrier Needs

Barrier needs refer to overcoming technological barriers, particularly those faced by individuals who are less familiar with digital tools, as well as linguistic and cultural barriers arising from different cultures and religions. Suggested functionalities that could resolve such needs include: (i) the design and development of an inclusive interface with multilingual support, a voice assistant, virtual coach sessions, real-time assistance, and translations services and (ii) the provision of training and educational content for health workers, patients, and caregivers to facilitate their the use of technological devices.

### 3.3. Selected Use Case for System Implementation: Data Gathering, Sharing, and Planning

Following the assessment of emerging needs and the subsequent analysis of the desired outcomes and functionalities, three distinct potential use scenarios for the smartHUB approach were identified:Collection of medical data for remote health monitoring, including data sharing among healthcare professionals and support through a virtual coach;Collaboration among FCNs to efficiently manage patients in accordance with the geographical proximity of the healthcare professional to the patient and traffic conditions in urban areas;A multilingual video interpretation service to facilitate communication between patients and healthcare providers from different cultural backgrounds.

Considering the identified necessity of appropriate ICT systems customized to the specific operational demands of healthcare workers involved in territorial care management, encompassing the creation of a comprehensive patient monitoring history, alongside the well documented increase in life expectancy and the importance of domiciliary care, we decided to prioritize the first scenario. In response to the emerging challenges arising from the exponential growth in the number of patients receiving home-based care, we proposed the development of the PROASSIST 4.0 system with the purpose of ensuring real-time personalized management of patients for 4.0 territorial care. Specifically, the PROASSIST 4.0 system primarily aims to (i) delineate dynamic and adaptive services capable of addressing the needs of citizens and supporting healthcare professionals in managing a progressively growing patient population through a 4.0 territorial care approach, exploiting the potential of ICT solutions by integrating them with the care models already employed in healthcare and (ii) ensuring efficient planning of territorial care in accordance with occurrences and the actual needs of patients and caregivers. With the introduction of these tools, the PROASSIST 4.0 system intends to align with new paradigms for complex organizations, requiring flexibility, adaptability, and optimization, and to assure the maintenance and promotion of the well-being and safety of all individuals involved in the process.

## 4. PROASSIST 4.0 System Architecture Overview

Figure 3 provides a comprehensive overview of the PROASSIST 4.0 system’s architecture, highlighting its main components and the interconnections and communication protocols used for data exchange.

As detailed in the following sections, the PROASSIST 4.0 system facilitates the collection, processing, and storage of health data entered by patients and FCNs. Data can be manually inserted or acquired by wireless wearable sensors used for clinical parameter measurements and monitoring. Through the PROASSIST 4.0 app, patients can independently input their health data and access educational content and tutorials to manage their health effectively, fostering self-monitoring. Additionally, nurses are allowed to input health measurements taken during home visits, thereby enhancing home care health services and promoting a more comprehensive patient care.

Privacy and security are paramount. The PROASSIST 4.0 system has been developed in full compliance with GDPR regulations. During the preparation of the experimental phase, a Data Protection Impact Assessment (DPIA) has been conducted alongside a comprehensive risk analysis, as part of the study protocol submission to the ethics committee. Data entered by users are securely transmitted to the cloud server via the HTTPS protocol for storage in the platform’s cloud-based database. These data undergo basic processing, including organization, generation of historical data in graphical formats, and the creation of filtered views tailored to individual user profiles. The cloud deployment of the platform is hosted on Amazon AWS, ensuring compliance with the General Data Protection Regulation (GDPR), AgID certification, and servers located within the European Union. To minimize the risk of data loss, stored data undergo periodic backups. The user profile structure allows patients to access only their own entered data, while nurses can access the data of their assigned patients, who have provided informed consent and received proper training during the enrollment phase. The data undergo biweekly backups to ensure protection and availability. Furthermore, transparent privacy policies and consent procedures are clearly communicated to users, building trust and confidence in the app’s use.

Notably, the PROASSIST 4.0 system has been specifically designed from its inception to be modular, scalable, and open to integration with other platforms. The interoperability is crucial in the healthcare context to enable a multidisciplinary approach by allowing different healthcare professionals and operators to manage patient care collaboratively. Custom APIs can be developed to facilitate seamless integration of the PROASSIST 4.0 system with other healthcare management platforms and external applications.

The PROASSIST 4.0 architecture is composed of the following interlinked macro-modules:Web front-end;Web back-end;A mobile application (IOS and Android);

The web front-end is accessible via a desktop platform by the system administrator for management of users (both patients and nurses). The creation of profiles and the subsequent assignment of patients to their respective nurses allow patients to access and view their own medical data and nurses to retrieve the data of their patients by the mobile application. All the data processing, management, and exchange based on specific permission is provided by the system back-end hosted in the cloud (PROASSIST 4.0 IT Platform).

In order to give an overview of the system functionalities, in Figure 4, the high-level functional diagram of the PROASSIST 4.0 system is shown, highlighting both the end-users’ side and the IT platform side. More detailed analyses of end-users’ side’s functionalities are carried out in the following section, where the macro-functionalities of the PROASSIST 4.0 app are described.

## 5. PROASSIST 4.0 App

The PROASSIST 4.0 app constitutes a digital tool for the collection and sharing of information regarding the patient’s health status. Data gathering occurs either through self-monitoring by patients in their daily life or during home visits by FCNs, who provide patient assessments, evaluations, and care. In the former case, the app aims to enhance patients’ awareness and management of their health, thereby encouraging their active participation in the care process. In the latter case, the PROASSIST 4.0 app supports home-visit nurses in delivering home care services. A real-time remote monitoring system enables the collection of patients’ health parameters, the identification of unexpected critical issues, and the adjustment of nurses’ visit scheduling to ensure timely care based on emergent priorities.

### 5.1. Macro-Functionalities

The PROASSIST 4.0 app provides the following key functionalities, accessible through an internet connection:Profile creation and management: the app allows the creation and management of users’ profiles, currently limited to patients and nurses, with future expansions planned to include other healthcare professionals.Health data acquisition and visualization: users can input and view various health data, including vital signs and medical reports, with both nurses and patients able to monitor this information through textual and graphical formats.Communication: the app facilitates one-way communication from nurses to patients, with plans for future updates to support two-way interactions.Information and training: a “Resources” section offers video tutorials and technical support, ensuring users are well-informed about both the app’s usage and health-related topics.

### 5.2. Profiles and Related Functionalities

Currently, the PROASSIST 4.0 app features two distinct profiles designed for patients and nurses (Figure 5). The patient profile enables users to input, view, and share their health data, while the FCN profile offers tailored functionalities for the monitoring and management of patient care. In response to feedback from nurses, a patient care planning function has been recently integrated into the nurse profile, aligning with the smartHUB approach. As a result, the app now supports self-monitoring for patients and professional monitoring along with patient care planning for nurses. Below is a more detailed description of these functionalities.

#### 5.2.1. Patient Self-Monitoring

The PROASSIST 4.0 app offers various functionalities to support patients in self-monitoring. Patients can enter real-time health measurements, such as blood pressure, heart rate, body temperature, and more, either manually, via voice input, or through transcription. Color-coded alerts notify patients if entered values fall outside physiological ranges, ensuring immediate awareness of potential errors (Figure 6). Self-monitoring also includes subjective pain levels and the recording of medical reports, such as pulmonary function tests and electrocardiograms. Additionally, patients can report data on the management of wounds and medical devices, either independently or with caregiver assistance. The app allows patients to view a graphical history of their self-reported data with time filters and provides notifications when no measurements have been recorded for the day. Access to caregiver and healthcare professional contact information is also available, empowering patients to manage their health proactively.

Nurses play a crucial role in facilitating patient adoption and effective use of these self-monitoring technologies [32]. By guiding patients, educating them on consistent monitoring, and fostering collaborative relationships, nurses help ensure patient engagement and adherence to care plans [33].

#### 5.2.2. Professional Monitoring

The PROASSIST 4.0 app offers a wide range of features tailored to the FCN profile. Upon logging in, nurses can view a personalized list of their assigned patients and access individual profiles and check-ups. Visual notifications highlight unread check-ups, changes in patient complexity, and the number of care activities scheduled for the day.

During patient check-ups, nurses can enter medical measurements and reports in real-time, either manually, via voice input, or transcription. These check-ups extend beyond basic metrics to include notes on oxygen therapy, stoma or tracheostomy presence, palliative care activation, and the management of pressure injuries, wounds, and medical devices. Nurses also assess the patient’s overall health, social context, and autonomy status. Interactions with other healthcare providers are documented as well. Historical data can be viewed in graphical format, with filters for the data source and time range (Figure 7). The app also provides access to patient and caregiver contact information.

#### 5.2.3. Patient Care Planning

In addition to enabling nurses to collect critical information for monitoring patients’ health, the PROASSIST 4.0 app facilitates comprehensive patient care management through its patient care planning feature. This tool allows healthcare professionals to create a periodic schedule of all necessary care activities. By mapping the patient care journey, the app provides an overview of completed tasks, planning activities, and any gaps in care.

Nurses can design personalized care pathways for each patient by scheduling required tasks on a calendar, specifying times and frequencies. The daily program for each patient is modifiable and can be interrupted as needed. Nurses can view detailed schedules for the current and upcoming weeks, with planned activity days highlighted. Each patient’s card shows scheduled tasks for yesterday, today, and tomorrow, along with non-postponable activities.

During home visits, nurses record whether tasks are completed, not completed (with reasons), or marked for future completion. Unperformed activities can be rescheduled according to the patient’s needs and nurse’s workload. Color-coded notifications indicate delayed tasks, with the severity of the delay noted (Figure 8). These features aim to streamline care planning, reduce costs, minimize travel time, balance workload, and maximize the number of visits while considering patient needs.

## 6. Experimental Activities

This section provides an overview of the experimental activities conducted within the framework of our approach. The first subsection details the results from a past experiment, while the subsequent subsections outline activities planned for the future:Preliminary results for system tuning;Upcoming large-scale validation of the PROASSIST 4.0 system;Future validation of patient care planning functionality.

### 6.1. Preliminary Results for System Tuning

Our initial findings regarding the use of the PROASSIST 4.0 app arise from the fourth workshop with FCNs (Figure 2), during which the initial co-designed version of the app was introduced and assessed. We aimed to collect preliminary feedback on the app’s professional monitoring functionality, which is essential for subsequent system refinements. The preliminary system tuning was conducted to gather feedback from nurses before proceeding with validation in real operational contexts, which will involve the structured collection of data that will be analyzed to provide both quantitative and qualitative insights into the system’s performance. Early results suggested that, despite still being in a development and refinement phase, the PROASSIST 4.0 app demonstrated an overall good functionality. Specifically, for the majority of nurses involved, the app achieved a good level of usability, with a small percentage of users rating it as excellent. However, some users expressed that there is still potential for improvement in app usability (Figure 9). Additional results of this phase are reported in [31].

Moreover, other results that helped enhance the app usability were derived from the analysis of the collected feedback on areas for improvement. The desired features identified through discussions with nurses are detailed in Table 1. It is worth noting that the table presents a synthesis of the results derived from the analysis of all the feedback gathered from nurses during the workshop and subsequent iterations to better define the suggested features or improvements. Some of them (i.e., voice data entry, assistance, resources, and in-app questionnaire) have already been implemented into the latest release of the app, which will be used in the next testing phase involving both nurses and patients. For other features (i.e., nurse-to-nurse patient transfer, offline use, system integration and interprofessional communication, and multilingual accessibility), development is planned for the future.

We expect that refining the system based on feedback from nurses will significantly enhance app usability, especially when evaluated in a real operational context—as will be carried out during the upcoming validation of the PROASSIST 4.0 system—rather than during a single workshop session, where time constraints may not afford a comprehensive understanding of the app’s true usability.

### 6.2. Upcoming Large-Scale Validation of PROASSIST 4.0 System

The first phase of future experimental activities encompasses a validation study that will have a total duration of 60 days. This constitutes a prospective observational pilot study aimed at refining the functionalities of the health digital app we proposed, which, to date, has been developed exclusively within a theoretical testing environment. The final product will serve as a support for healthcare providers and caregivers in assisting patients at their home, and will also be available through territorial services to all those who require it. Specific objectives of this validation study will be as follows:To assess acceptability and usability of the PROASSIST 4.0 app in a real-life operational context that involves patients receiving community care services and home-visit nurses;To analyze the feedback collected from patients in order to refine the basic functionalities and enhance usability of the app, thereby increasing acceptability among end-users.

Approximately 120 patients and 20 FCNs of LHU Toscana Centro will be recruited on a voluntary basis. Patients will be evaluated and enrolled within their living environment during their scheduled home visits as part of their care plan. Inclusion criteria will include: (i) being under the care of the identified FCNs district; (ii) having a smartphone or tablet with iOS 15+ or Android 8+ and an internet connection; (iii) having familiarity with smartphone and app use; (iv) being helped by a caregiver capable of entering data in case of patients with limited digital skills; and (v) being at least 18 years old. Enrolled patients will provide written informed consent and will be requested to use the app to input their health data throughout the study period while continuing to receive their standard care treatments according to their care plan. It will be emphasized that the experimentation is purely functional and lacks any clinical care aspect. Specifically, it will be highlighted that health data entered by patients will not undergo a clinical evaluation, and any anomalies identified during the measurements should be directed to competent healthcare professionals.

To ensure effective use of the system, both FCNs and patients will receive initial training and ongoing support, which are crucial for successful technology adoption. Prior to the start of the study, nurses will be trained on how to use the application and how to instruct patients to use it independently. During patient enrollment, which will be conducted by the FCN during home visits, the nurse will provide the patient with a prototype of the PROASSIST 4.0 app to install on smartphone or tablet, along with a detailed explanation of its functionality. The FCN will demonstrate the use of the system and address any questions or concerns. Following this demonstration, the patient should be capable of independently performing measurements according to the provided instructions. If any issues arise during the study, the patient can contact the assigned nurse, who will conduct an initial assessment of the problem. Depending on the nature of the issue, the nurse will either provide direct support or request technical assistance. Both nurses and patients will have access to ongoing technical support throughout the study, with the specific features of this support detailed in Table 1.

The PROASSIST 4.0 app functionalities that will be evaluated in this phase are patient self-monitoring and professional monitoring by FCNs. These two features constitute the core capabilities of our app, as they allow health measurements and data monitoring for both patients and nurses. The patient care planning functionality represents a highly customized feature that directly responds to specific healthcare needs and extends beyond basic functionalities. Therefore, this feature will be separately tested in a later phase.

To assess the acceptability and usability of the PROASSIST 4.0 app for both patient and nurse interfaces, specific variables will be evaluated: total number of measurements, total measurements per patient, total measurements per parameter, total measurements per parameter per patient, number of support requests, and total registrations. System performance metrics, including uptime, robustness, and response times, will be extracted directly from the PROASSIST 4.0 platform. These metrics are largely influenced by the server provider (Amazon AWS) and the system configuration. System resources can be scaled as needed based on continuous monitoring of the system load. Additionally, mechanisms for anonymously tracking end-user behavior within the app have been implemented. All these parameters will enable us to examine user behavior and interface usability. We expect that the interface of our mobile application will receive positive feedback from users, given their active involvement in its creation. The collected data will be analyzed in conjunction with information gathered from an anonymous in-app questionnaire, which both patients (or caregivers) and nurses will complete during the experimentation phase. The following information will be collected through the questionnaire:Sociodemographic characteristics of app users;Digital literacy of app users, to evaluate their level of knowledge about the use of smartphone and tablet in daily life;App usability, assessed through the System Usability Scale [31,34,35];App functionality, evaluated through technical indicators specifically developed to assess specific features of the application, such as data collection, health monitoring, and data visualization.

The questionnaire will be completed in a pseudonymized manner using an alphanumeric code. As previously described, all data will be processed with consistency and responsibility, ensuring full compliance with the GDPR. Information collected will be utilized to evaluate the ease of use, comprehensiveness, and accuracy of the PROASSIST 4.0 app. This will support app developers in addressing any issues encountered during the experimentation phase and in the ultimate refinement of the system.

### 6.3. Future Validation of Patient Care Planning Functionality

Given the recent advancements in the development of our app with the implementation of the patient care planning, and recognizing the importance of this feature for FCNs, we are planning to conduct a series of experimental sessions focused on evaluating both the usability and effectiveness of this new feature. As in previous phases, users will receive appropriate training and technical support for using the system.

This preliminary validation will provide valuable insights and feedback, which will be pivotal for the ongoing refinement of the app. The information gathered will play a crucial role in guiding enhancements to ensure that the app effectively meets the real needs of healthcare assistance management.

## 7. Discussion and Challenges

In this study, we introduced the smartHUB approach, an innovative multidisciplinary systemic methodology designed to address the specific needs of community healthcare assistance of LHU Toscana Centro and ultimately enhance patient care and quality of life. Within the scope of the smartHUB project, the PROASSIST 4.0 system has been developed to provide real-time personalized patient management for advanced territorial care. Specifically, the PROASSIST 4.0 system includes a mobile application named the PROASSIST 4.0 app, designed for the collection and sharing of medical information regarding patients’ health. This app is currently available for both patients and nurses, providing self-monitoring functionalities for patient, as well as professional monitoring and patient care planning functionalities for nurses.

### 7.1. Strengths and Impact of Our System

To the best of our knowledge, this is the first study to describe the implementation of a bottom-up approach aimed at improving the management of community healthcare assistance. While previous studies have proposed bottom-up strategies in the context of digital health solutions and chronic care management [13,14,16,18,19,20], none have specifically addressed the home-visit nursing setting.

A key distinction of our approach is the extensive and well-structured preliminary analysis of end-users’ needs that we conducted. Although some studies have employed a participatory design approach that includes all healthcare professionals involved, including nurses, they often lacked such a comprehensive assessment of healthcare needs [16,18]. Moreover, most of these studies focused primarily on designing ICT systems that enable patients to monitor their own health status, rather than taking a broader view of the healthcare ecosystem. Despite the participation of nurses in the design process of these platforms, they appear to be more oriented toward facilitating patient self-monitoring rather than supporting nurses in managing their daily workloads [13,16,18,20]. As a result, these platforms tend to position patients as the primary end-users, limiting their utility for nurses. In contrast, our model not only involves nurses in the design process but also explicitly positions them as key end-users of the developed system. This ensures that the platform provides practical support for nurses in their daily activities. Additionally, many previous studies have been disease-specific, targeting cohorts of patients with particular clinical conditions [13,14,15,16,17,18,20]. Mobile health solutions are frequently tailored to specific medical conditions [36,37,38], which limits their adaptability to other diseases or broader healthcare settings. On the contrary, our approach offers a comprehensive methodology designed to support community healthcare management on a larger scale, delivering a high-level solution adaptable to various contexts beyond specific medical conditions.

To provide an overview of the main impact of our study, we have analyzed the strengths and opportunities related to the proposed system. Among its key strengths are the promotion of new health initiatives and the implementation of the FCN model, which emphasizes patient-centered care. Additionally, the system incorporates a dedicated digital health record for nursing activities, facilitating more efficient data management and increasing healthcare professionals’ skills. The opportunities offered by our system include improved health status monitoring, enhanced data sharing capabilities, an alert system for early intervention, and greater patient compliance with therapy. As part of our strategic approach, we propose testing and validating the system with end-users who were directly involved in its conceptualization and development, in line with our co-design methodology.

As previously described, a fundamental aspect of our approach deals with the early engagement of end-users in the developmental of the final solution, from the initial phase of identification and analysis of needs and extending throughout the entire design process. This critical aspect of our methodology stems from the understanding that continuous involvement of end-users in the development and implementation stages is essential for identifying factors that influence app acceptability and practical use [39,40]. For example, insufficient app usability can adversely affect the adoption of the technology and potentially compromise patient health. Conversely, enhanced usability is closely associated with user satisfaction, leading to broader adoption of the tool, which can significantly improve community healthcare assistance and the quality of life of patients [41,42]. Thus, the development of the PROASSIST 4.0 app is guided by users’ feedback to ensure the optimization of app ease of use. Although initial feedback from nurses revealed that not all end-users were fully satisfied with app functionality [31], we are confident that our ongoing engagement with nurses and patients will result in improvements that will enhance app utilization and usability in future assessments. Targeted training designed to raise community awareness about the benefits of the customized 4.0 approach will also contribute to foster active participation and ownership of the proposed methodology in both patients and nurses, which is essential for its success. Furthermore, adequate training and support programs will be crucial to empower end-users with the necessary digital literacy and skills to effectively utilize ICT tools.

### 7.2. Challenges and Barriers to Adoption

Successfully implementing the system requires acknowledging and addressing the complex challenges and barriers it presents. Key issues include interoperability among various systems, data security, and infrastructure limitations [32]. Interoperability is critical for the seamless exchange of patient information across different platforms. Without it, the coordination and continuity of patient care can be compromised. Equally important are robust data security measures to protect sensitive patient information from breaches and cyber threats. Electronic health records are particularly susceptible to cyberattacks, highlighting the need for strong encryption and access controls to maintain data privacy and security, which remains a major concern for the adoption of technological solutions [43]. Furthermore, inadequate infrastructure, such as outdated hardware, limited bandwidth, or unreliable internet connectivity, can significantly impede the efficient operation of ICT systems. These limitations may cause delays in accessing patient data or lead to system crashes, thereby negatively impacting nurses’ ability to provide timely care. Addressing these technical challenges necessitates concerted efforts in standardization, significant investment in robust IT infrastructure, and ongoing technological advancements.

The long-term adoption of the proposed system may face barriers such as different levels of technological literacy and resistance to change from both healthcare providers and patients. Not all nurses or patients have equal access to technology or digital skills necessary for effective ICT engagement. In this regard, nurses are crucial in bridging the digital divide and ensuring inclusive care. Moreover, resistance to change may arise as patients struggle to integrate new technologies into their routines and healthcare professionals worry about increased workload or workflow disruptions. To overcome these challenges, fostering a culture of innovation, interdisciplinary collaboration, and providing comprehensive training and ongoing support are essential. Additionally, seamless integration of new ICT systems into existing workflows is necessary to improve efficiency and minimize disruptions to patient care routines.

### 7.3. Ethical Implications

Ethical complexities are closely interlinked with technical considerations [32]. Maintaining patient privacy and confidentiality remains a paramount concern in the digital era. Nurses face the delicate challenge of balancing the use of patient data to enhance care outcomes with respecting individual autonomy and privacy rights. To mitigate the risks of data misuse and violations of trust, it is imperative to rigorously adhere to ethical frameworks and guidelines on data sharing and patient consent. In our case, ethical considerations are safeguarded through the involvement and consultation of healthcare professionals at every stage of the design and development process, ensuring the creation of a high-quality system that meets the needs of its end-users. Concerning privacy and confidentiality, access to patient information displayed in the app is restricted to authorized users, specifically, licensed healthcare professionals and administrators of the platform, and is limited to their designated roles. Regarding patient consent, all home-visit nurses involved in the large-scale validation of the system will be thoroughly informed about the study and provided with detailed information and a patient consent form.

## 8. Conclusions and Future Work

In conclusion, we presented the development of an ICT platform called PROASSIST 4.0, specifically designed to enhance community healthcare management of LHU Toscana Centro. By adopting the smartHUB multidisciplinary methodology, which prioritizes the creation and implementation of ICT tools based on end-users’ needs, we developed a highly customized app that integrates patient monitoring (including self-monitoring), real-time health data collection and processing, patient care planning, and decision support for healthcare professionals. This solution not only increases patients’ awareness of their health status, making them active participants in their care journey, but also supports nurses in delivering patient-centered care. Our work highlights the importance of addressing healthcare needs and challenges with innovative solutions aimed at fostering a more connected and efficient healthcare environment.

Finally, future efforts will focus not only on the development of new features that effectively address users’ needs but also on the integration of innovative technological solutions for remote health monitoring. Various data collection systems, including wearable sensors, may be employed to monitor and record real-time data on physiological indicators and physical activity levels [44]. Furthermore, the implementation of advanced algorithms for data correlation (biometric parameters, physical activity, and therapy adherence) has the potential to significantly improve patient care by enabling continuous monitoring of critical health metrics, offering immediate insights into the measured data and allowing for the early detection of any anomalies [45]. This would facilitate more accurate and timely interventions. Such integrations would not only empower patients with greater control over their health but also enable healthcare providers to deliver more personalized and proactive care programs.

## Figures and Tables

**Figure 1 sensors-24-06059-f001:**
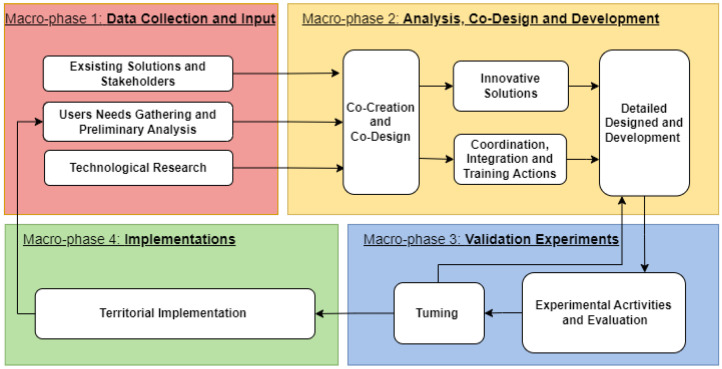
Macro-phases of the smartHUB multidisciplinary systematic approach.

**Figure 2 sensors-24-06059-f002:**
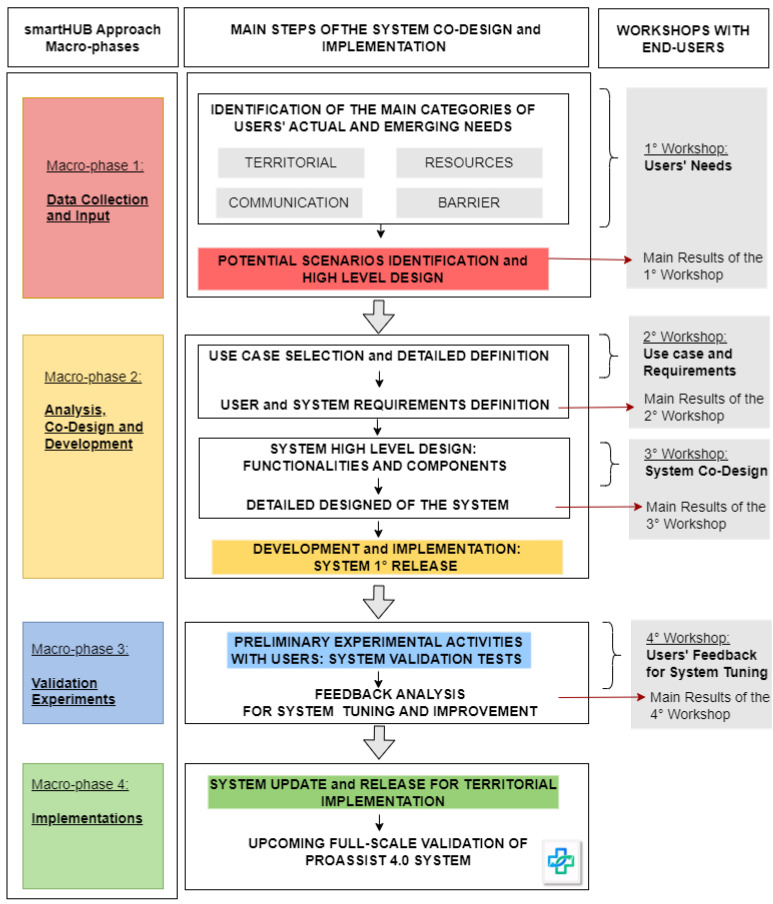
The smartHUB approach adoption: from users’ needs to PROASSIST 4.0 system.

**Figure 3 sensors-24-06059-f003:**
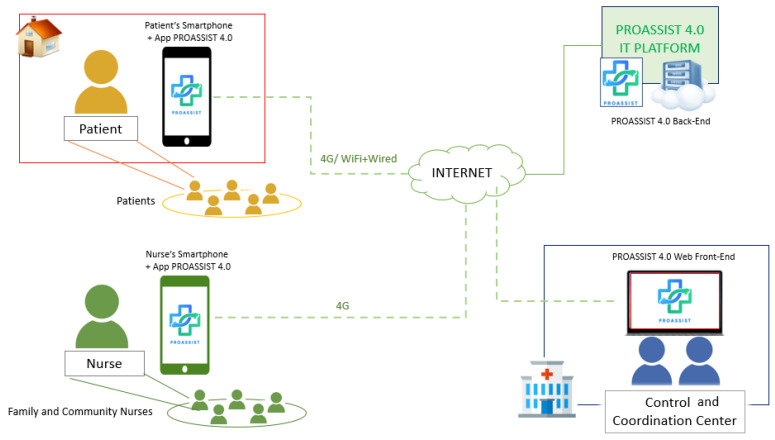
PROASSIST 4.0 system: architecture overview.

**Figure 4 sensors-24-06059-f004:**
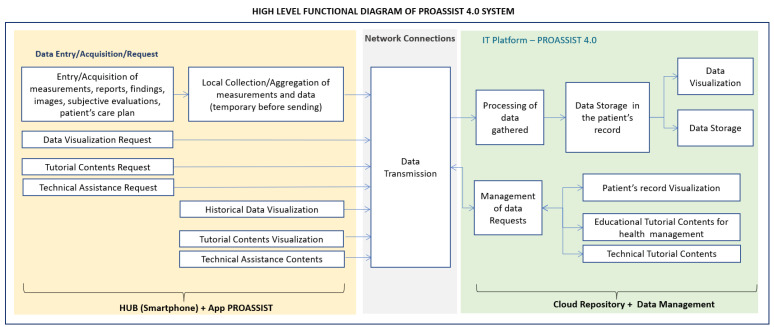
PROASSIST 4.0 system: high-level functional diagram.

**Figure 5 sensors-24-06059-f005:**
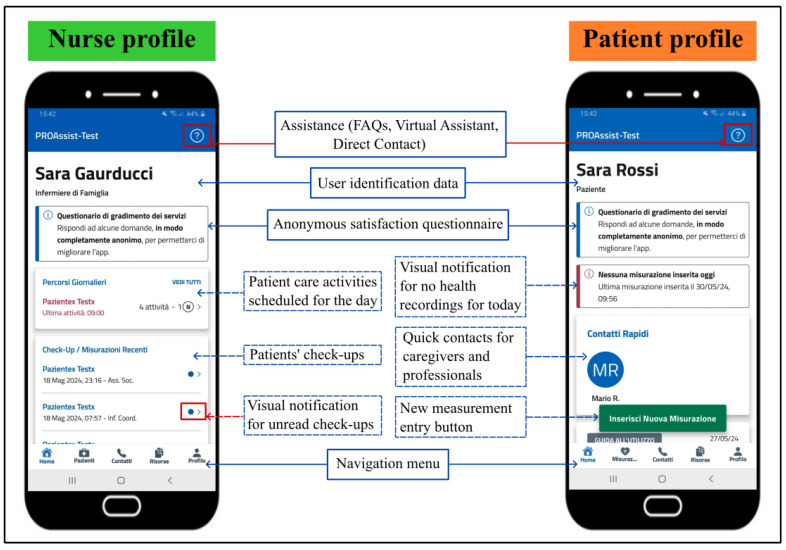
PROASSIST 4.0 app home-screen: FCNs and patient profiles.

**Figure 6 sensors-24-06059-f006:**
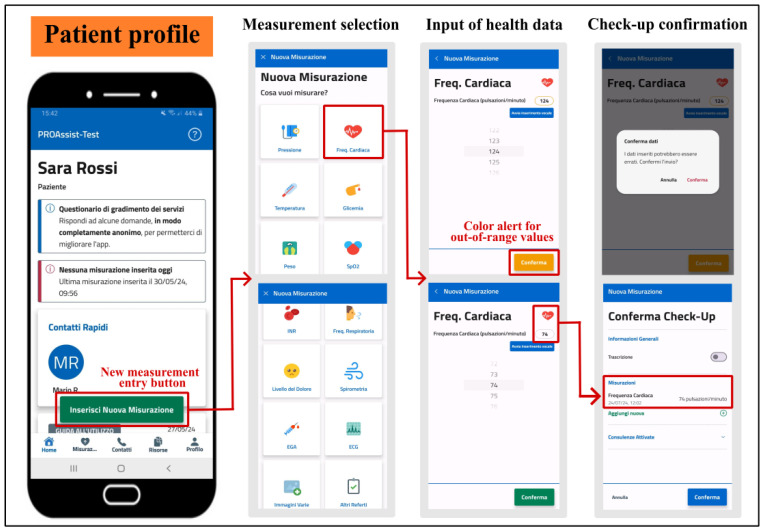
Entering a health measurement in the PROASSIST 4.0 app using the patient profile. The procedure is analogous to the FCN profile.

**Figure 7 sensors-24-06059-f007:**
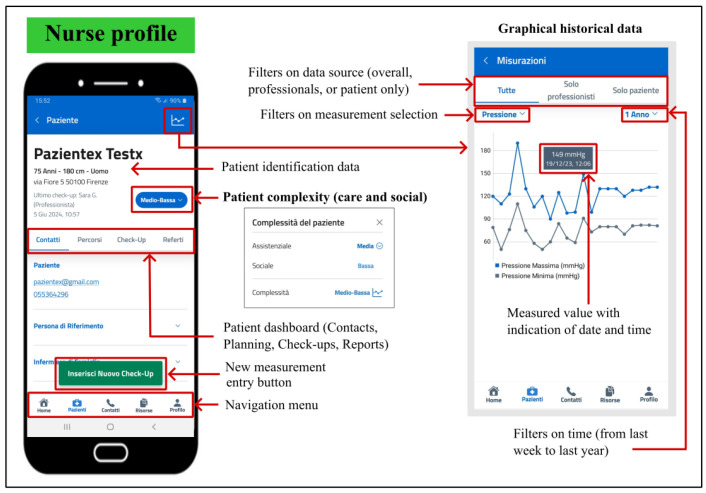
Patient card displaying graphical historical data for the FCN profile in the PROASSIST 4.0 app. The visualization of the measurement history is analogous to the patient profile, with the exception that patients are allowed to view only the measurements they have personally recorded.

**Figure 8 sensors-24-06059-f008:**
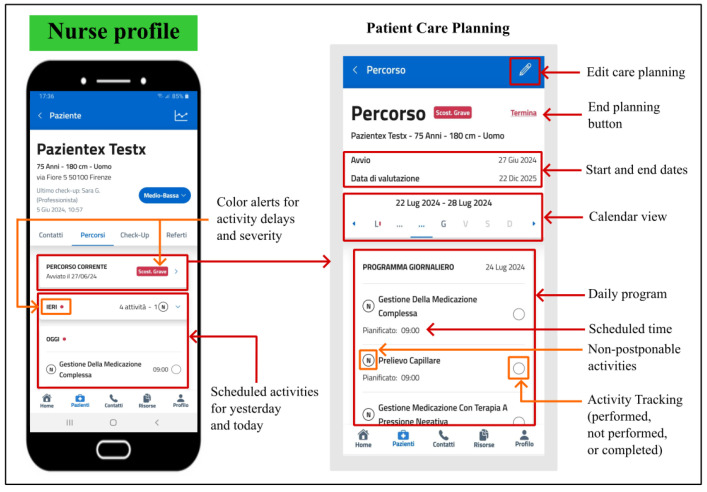
Patient care planning functionality in the PROASSIST 4.0 app for the FCN profile.

**Figure 9 sensors-24-06059-f009:**
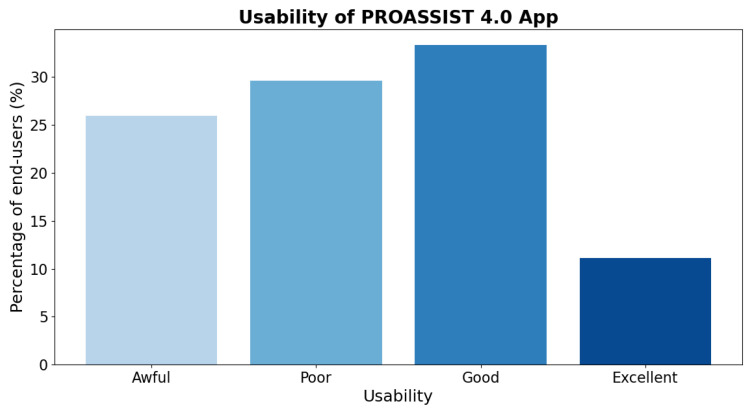
Preliminary usability of PROASSIST 4.0 app: percentage of end-users rating usability as awful, poor, good, or excellent.

**Table 1 sensors-24-06059-t001:** System tuning feedback: Features to be developed based on areas for improvement suggested by nurses during preliminary testing of PROASSIST 4.0 app.

Features	Description	Status
Voice data entry	A feature enabling both patients and nurses to input health data via voice commands has the potential to significantly increase app usability and streamline healthcare processes, offering a more user-friendly and efficient tool for managing health information.	Implemented
Technical support	Support resources represent a robust support framework designed to facilitate the effective use of the app. This includes a comprehensive FAQ section, organized separately for patients and nurses, addressing topics such as app functionalities, security and privacy, as well as measurements and check-ups. Additionally, a virtual assistant, accessible through a chatbot, provides real-time guidance and direct contact with support personnel, ensuring users receive immediate help and can navigate the app efficiently.	Implemented
Resources	Video tutorials on app usage and health measurement management are provided to assist users in effectively utilizing the system. Many of these resources are tailored specifically for both nurses and patients, ensuring that the information is relevant and targeted.	Implemented
In-app questionnaire	Users have highlighted the advantage of an in-app questionnaire over completing a manual form. With options for voice input, this anonymous questionnaire evaluates app usability and functionality, offering valuable insights into user experience and areas for improvement.	Implemented
Nurse-to-nurse patient transfer	Currently, nurses can modify only the health records of their assigned patients. This feature would enable the Nursing Coordinator to grant real-time temporary access to other patients’ records, allowing nurses to intervene when the assigned nurse is temporarily unavailable. It would also enhance collaboration among FCNs, particularly for real-time teleconsultations.	Planned
Offline use	This feature aims to ensure service efficiency and continuity of care even in areas with no network coverage. Users will have the capability to download patient data and historical records before a home visit. Data entered offline will be synchronized with the central system once network connectivity is restored, and all patient data will be subsequently removed from the device.	Planned
System integration and interprofessional communication	A key feature requested by users is the integration of the system with existing management platforms, and the communication and data sharing among all healthcare professionals involved in patient care. This would ensure a more comprehensive and collaborative approach to managing patient health.	Planned
Multilingual accessibility	By implementing a multilingual interface, the system aims to enhance accessibility and inclusivity for individuals from diverse linguistic backgrounds, thereby improving overall patient experience and optimizing the delivery of healthcare services.	Planned

## Data Availability

Data are contained within the article.

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
