# Peer review of "Health Community 4.0: An Innovative Multidisciplinary Solution for Tailored Healthcare Assistance Management"

_sensors, 2024, doi:10.3390/s24186059_

Round 1
Reviewer 1 Report
Comments and Suggestions for Authors
The paper presents a well-thought-out and innovative solution for improving community healthcare management. The work could be even more impactful with some enhancements, particularly in validation, scalability, technical details, and user experience. The authors are encouraged to provide more empirical data and insights into the system's scalability and user engagement strategies in future iterations. My suggestions to the authors for improvement of work are:
1. Evaluation and Validation:
While the paper describes the validation phase of the smartHUB approach, it would benefit from more specific details on the results of these validation experiments. For instance, quantitative data on system performance, user satisfaction, or healthcare outcomes would strengthen the argument for the system's effectiveness.
Including a comparison with existing healthcare management systems would also provide context and demonstrate the advantages or areas where PROASSIST 4.0 offers superior performance.
2. Scalability and Generalizability:
The paper discusses customizing the solution for different territorial needs. Still, it would be valuable to elaborate on how the system can be scaled to larger or different healthcare settings beyond the LHU Toscana Centro. It would also be beneficial to discuss potential challenges in scalability and how they might be addressed.
Additionally, exploring the system's adaptability to different cultural or regulatory environments could enhance its generalizability.
3. Technical Specifications:
The paper could benefit from a more in-depth discussion of the PROASSIST 4.0 system's technical architecture. While the high-level design is covered, more detailed technical specifications, such as data security measures, system interoperability, and performance metrics, would provide a clearer understanding of the system's robustness.
4. User Experience and Training:
The mobile application and platform's user experience (UX) aspects are mentioned, but further details on how UX research influenced the design would be useful. Including user feedback on the interface and how it was incorporated into the final design would make the case for the system's user-friendliness stronger.
The paper should also address the training provided to users, particularly healthcare providers, to ensure effective use of the system. This includes initial training and ongoing support, which are critical for technology adoption in healthcare.
5. Future Work and Sustainability:
The conclusion could be strengthened by discussing future work, particularly regarding system upgrades, potential integration with other healthcare technologies, and long-term sustainability. Addressing how the system will evolve to meet future healthcare challenges would add depth to the paper.
Reviewer 2 Report
Comments and Suggestions for Authors
This paper proposes a new strategy for the development of community healthcare improvement by employing a multi-disciplinary approach. The research focuses on the implementation of ICT in a patient oriented care approach that is of bottom up perspective. To the greatest extent, the most important achievement is the creation of the PROASSIST 4.0 evolution which contains a mobile application that can be tailored for assistance in the collection of health information, supervision of the patient and care management by the health providers. The proposed system follows the recent progress witnessing in the field of digital health. A methodological design called smartHUB was used, which is multidisciplinary in nature and was used in the course of the project involving all key players. This addresses that, the developed offering is needs based and user driven kind of solution. The contents provided under the 'architecture and operability of PROASSIST 4.0' are clear and functional as they highlight, thoroughly, the systems’ operability and the integration into the healthcare systems. Further, the manuscript also covers preliminary usability testing, which is important towards system development. The foreseen further validation studies are logically planned and reflect authors’ strive towards further solution enhancement.
Nevertheless, there are aspects that could be enhanced. Some sections, particularly those detailing the methodology and system functionalities, are unduly detailed. While the manuscript demonstrates thoroughness, greater conciseness in these areas would enhance reader engagement. It is recommended that the technical descriptions be summarised and that greater focus be placed on the impact and implications of the proposed solution. The background section provides a comprehensive overview of the context, but it could be enhanced by a more critical analysis of existing solutions. A comparison with analogous ICT-based healthcare systems would serve to reinforce the argument for the novelty and necessity of the proposed system. Incorporating contemporary developments in health informatics and the IoT in healthcare would enhance the paper's position within the existing body of knowledge.
The preliminary results presented are limited to the initial usability tests conducted with healthcare professionals. A more comprehensive evaluation of the system's effectiveness would be facilitated by the presentation of quantitative data or the provision of more detailed qualitative feedback. The forthcoming full-scale validation studies are of great importance; however, the manuscript should provide a more explicit outline of how these future studies will address any potential limitations or challenges identified in the preliminary results. Although the manuscript mentions the importance of data security and patient privacy, it lacks a detailed discussion on the ethical implications of using such a system, especially concerning data handling and patient consent. Expanding this section would provide a more robust framework for the ethical deployment of the system.
It would be beneficial for the manuscript to further explore strategies to ensure the sustained user engagement and adoption of the PROASSIST 4.0 system. It would be beneficial to address potential barriers to adoption, such as technological literacy and resistance to change among healthcare providers and patients, in order to enhance the practical applicability of the proposed solution. In conclusion, the manuscript presents a promising solution to improve community healthcare management through an innovative, ICT-based system. While the study is well-conceived and thoroughly executed, there are areas where the manuscript could be refined, particularly in terms of conciseness, depth of literature review, and validation of results. With these improvements, the paper would make a valuable contribution to the field of digital health 😊
Round 2
Reviewer 1 Report
Comments and Suggestions for Authors
Dear Editor,
The author has significantly improved the manuscript and addressed all my previous concerns satisfactorily. The revisions have strengthened the quality and clarity of the work.
I am pleased to recommend accepting the manuscript for publication in your journal.
Best regards,